# Milling of Three Types of Thin-Walled Elements Made of Polymer Composite and Titanium and Aluminum Alloys Used in the Aviation Industry

**DOI:** 10.3390/ma15175949

**Published:** 2022-08-28

**Authors:** Krzysztof Ciecieląg, Kazimierz Zaleski

**Affiliations:** Department of Production Engineering, Faculty of Mechanical Engineering, Lublin University of Technology, 36 Nadbystrzycka, 20-618 Lublin, Poland

**Keywords:** milling, thin-walled elements, surface roughness, vertical force, deformation, aluminum alloy, titanium alloy, polymer composite

## Abstract

The machining of thin-walled elements used in the aviation industry causes may problems, which create a need for studying ways in which undesirable phenomena can be prevented. This paper presents the results of a study investigating face milling thin-walled elements made of titanium alloy, aluminum alloy and polymer composite. These materials were milled with folding double-edge cutters with diamond inserts. The results of maximum vertical forces and surface roughness obtained after machining elements of different thicknesses and unsupported element lengths are presented. The results of deformation of milled elements are also presented. The results are then analyzed by ANOVA. It is shown that the maximum vertical forces decrease (in range 42–60%) while the ratio of vertical force amplitude to its average value increases (in range 55–65%) with decreasing element thickness and increasing unsupported element length. It is also demonstrated that surface roughness deteriorates (in range 100% for aluminum, 30% titanium alloy, 15% for CFRP) with small element thicknesses and long unsupported element lengths. Long unsupported element lengths also negatively (increasing deformation several times) affect the accuracy of machined elements.

## 1. Introduction

The machining of thin-walled elements for the aviation industry has a significant impact on their surface structure geometry and deformation caused by cutting [1]. In machining thin-walled materials, we can observe a vast number of factors and problems that have a significant impact on the shaped elements. Among problems, we can distinguish excessive surface roughness, deformation of thin elements, vibrations occurring during machining or rapid wear of tools [2,3,4]. Problems related to the machining of thin-walled elements are analyzed by many researchers, and conclusions drawn from their research provide important guidelines for industry. Numerous studies have shown that great focus is put on the problem of milling. Thin-walled elements used in aviation are very often made of aluminum and titanium alloys and polymer composites.

The machinability of aluminum alloys is related to the cutting forces occurring during machining. Aluminum alloys are tough materials, which create a large contact area between the chip and the tool. A too long contact causes increased cutting forces and heat generation; however, taking into account low shear strength, the machining of aluminum alloys can be classified as a relatively simple technological process [5]. The addition of alloying elements to increase the mechanical strength and hardness of the tested aluminum alloy creates a scenario whereby, as the contact surface of the chip with the tool reduces, so the cutting forces can be reduced too. A study on aluminum alloys 7075-T6, 6061-T6 and 2024-T351 has shown that the 7075-T6 alloy, which has the highest strength and hardness, is characterized by the lowest vertical forces. Compared to the weakest material, 6061-T6, these forces were up to 30% lower [6].

Cutting forces are also related to the material and geometry of tools. The use of diamond-coated or fully diamond tools reduces cutting forces due to their high hardness and very low chemical affinity to aluminum [7,8,9]. Geometric changes caused by blade wear and built-up edge effect lead to increased cutting forces in machining aluminum alloys. Increased rake angle reduces the contact area of the chip with the tool, which reduces the cutting force [10]. Excessive cutting forces generated when machining with tools with small rake angles also contribute to greater flank wear [11].

Studies investigating the effect of the technological parameters of milling aluminum alloys on the cutting forces in this process have shown that increasing the cutting speed causes a decrease in the value of the cutting, forces regardless of the strength properties of the material [12,13]. The reduction in cutting forces when milling at high speeds (up to 450 m/min) in relation to the value of these forces when milling at low speeds (up to 150 m/min) can be as high as 25% [12]. However, the use of high cutting speeds in machining aluminum alloys is not recommended as it may increase the cutting temperature and toughness of the material [14]. On the other hand, increasing the feed speed significantly increases the value of the cutting forces, which for small feeds (0.05 mm/tooth) are several times smaller than for very high feeds, up to 0.2 mm/tooth [15]. Increasing the feed speed and the depth of cut in machining aluminum alloys causes shear difficulties, which leads to increased cutting forces [16,17].

The surface structure geometry of aluminum alloys obtained by machining depends on their hardness and microstructural properties [9,18,19]. Higher hardness results in lower surface roughness. This is because hardness reduces the material’s ability to stick to the rake face of the tool [1]. In machining aluminum alloys, high cutting speeds can cause surface deterioration due to rapid flank wear [20]. The surface roughness of aluminum alloys after milling also depends on the feed rate, because increase in this speed reduces the quality of the obtained surface [21].

The machining of thin-walled elements made of aluminum alloys poses problems primarily in the aviation industry, the most serious one being a lack of stiffness of the workpiece and its resulting deformation [22]. Stresses generated during this type of machining and the way it is performed have a direct impact on the use and properties of the element [1]. During the machining of thin-walled structures, bending stresses are of great importance [1]. The machining of thin-walled structures made of aluminum alloys is mainly carried out using numerically controlled machine tools that ensure the required accuracy of elements [23]. The wall thickness of a workpiece affects its deformation and surface roughness. It has been shown that doubling the thickness of a shaped element can lead to a twofold increase in its deformation [24]. Studies investigating the impact of milling parameters on the accuracy of thin-walled structures have shown that a threefold increase in the feed speed causes an over 50% increase in deformation and a 70% increase in surface roughness [2]. The second tested parameter was cutting speed, which—when increased—caused a two-fold reduction in surface roughness (when comparing low cutting speeds of about 80 m/min with the speeds up to 350 m/min). Increased cutting speed also led to reduced workpiece deformation, but only in the range up to about 250 m/min—above this value the deformation increased [2]. Other studies have also shown that higher geometric accuracy of the manufactured elements is related to an increase in cutting speed. High cutting speeds are recommended to minimize distortion of thin-walled components [25]. The third tested parameter, the depth of cut, did not have a significant effect on the deformation and surface roughness of produced parts [2]. The research on machining thin-walled structures sometimes provides contradictory results, because there are also studies showing that the thickness of an aluminum alloy element does not significantly affect the 2D and 3D surface roughness parameters [3]. Due to the accuracy of workmanship and deformation of thin-walled elements made of aluminum alloys, it is recommended to use carbide tools or tools with PCD blades [25].

In addition to aluminum alloys, titanium alloys are also used in the aerospace industry [26,27]. They are used for their properties such as low density, high strength, low Young’s modulus and high resistance to corrosion [28]. Low thermal conductivity and high hardness of titanium alloys make them difficult-to-machine materials, yet in many applications they replace steels and aluminum alloys.

The milling of titanium alloys is one of the most difficult operations due to high cutting resistance. High cutting resistance results from low thermal conductivity, high strength, high chemical reactivity and high stresses at the cutting edge [29,30].

For machining titanium alloys, cemented carbides are a widely used tool material; however, increased blade wear causes increased surface roughness [31,32,33]. Therefore, tungsten carbide tools are only used for low cutting speeds [30]. The most wear-prone area is the tool flank, and this wear can be reduced by the use of fine-grain carbide tools [34]. A solution to the rapid wear of tungsten carbide tools is the use of tools with CVD, PVD and PCD coatings [30].

The machining of titanium alloys is characterized by lower maximum technological parameters, especially when compared to the machining of aluminum alloys. The cutting speed of titanium alloys can reach 100–110 m/min, and the feed rate can be up to 0.1 mm/tooth [28,35], but the use of coolants is required to use high cutting speed and feed rates [36]. In machining titanium alloys, an increase in surface roughness is strongly related to an increase in cutting speed [35].

The milling of thin-walled titanium alloys is used to reduce the mass of manufactured elements [37]. A common problem in the machining of thin-walled titanium alloys is vibration because it affects process stability [4]. It has been shown that the vibration increases with cutting speed [4]. The machining of thin-walled elements is also associated with the problem of obtaining good surface structure geometry. Changing the method of support can lead to reduced deformation of components and lower risk of deformation when the components are clamped [38]. There have also been attempts to increase productivity by up to 40% through the use of large milling depths and the sandwich milling method [39].

Composites are an alternative to aluminum and titanium alloys used in the aerospace industry [40]. A significant group of this type of materials are carbon, glass and Kevlar fiber-reinforced plastics supersaturated with epoxy resin [41]. In many application areas the value of the ratio of tensile strength to specific weight is of vital importance. Carbon fiber-reinforced plastics are known to have the highest value of this ratio [42]. In addition to good tensile strength, these materials are also characterized by good corrosion resistance and high stiffness, which is the reason for their use in the construction of aviation components [43,44]. Due to a growing use of this type of materials, it becomes necessary to have sufficient knowledge about machining these materials.

For milling polymer composites, sintered carbides tools with polycrystalline diamond (PCD) inserts are used because they have high abrasion resistance while maintain sharp cutting edges for a long time [45]. Carbide (without PCD) and boron nitride (CBN) tools wear quickly in composite machining [46].

Cutting forces occurring in milling polymer composites are mainly related to the orientation and strength of the fibers [47]. Most studies devoted to cutting force prediction give similar results and conclusions. It has been found that this force primarily depends on the rake angle, fiber orientation and fracture toughness, as well as milling parameters [48,49,50,51].

The selection of parameters in the milling of polymer composites depends on the material structure, fiber orientation, fiber type and their share in the material [52]. In most cases, cutting speeds of 20 ÷ 250 m/min [46,53,54,55,56] and even 500 m/min [57] are used for milling carbon fiber-reinforced plastics (CFRPs). This type of material is usually processed using feeds ranging 0.01 ÷ 0.5 mm/tooth [50,53,54,57] and depths of cut from 0.1 to 4 mm [50,54,57,58,59]. Regarding milling of carbon fiber-reinforced plastics, it has been shown that feed rate has the greatest influence on the surface roughness of this material; meanwhile, the effect of cutting speed is slightly smaller and the influence of the depth of cut is insignificant. In most cases, surface roughness increases with feed rate [60].

Research on thin-walled composite materials is still a relatively recent issue. In the production process, composite materials are created when formed in special molds; therefore, most studies are focused on investigating the strength and damage of composite profiles by experimental techniques and the finite element method [61,62]. Composite materials are machined due to insufficient surface quality of parts after forming in molds. Scientific studies on machining composites have shown that the cutting force initially increases, and then decreases, with increasing cutting speed. The increase in cutting forces also results from increasing feed [63]. The results of milling thin-walled composite structures reported in [64] have also shown that the increase in feed rate causes cutting forces to increase in the entire range of the parameters applied. However, in contrast to [63], the analyzed cutting forces were observed to initially decrease and then increase for the applied high cutting speeds. The limited number of studies on milling thin-walled composite structures proves that this problem has not been yet fully explored.

The novelty of this study is that it investigates the milling process for thin-walled elements made of aluminum and titanium alloys and carbon fiber-reinforced plastic with epoxy resin. The milling of such materials is mainly used to obtain appropriate accuracy and surface roughness. The aim of the study is also to assess the machinability of three materials used in the aviation industry under the same milling conditions. The paper offers a comparative analysis of the machining process for thin-walled elements made of aluminum and titanium alloys and a polymer composite. In this study, the same technological parameters as found in milling are used to determine the effect of these parameters on cutting forces, surface roughness and accuracy resulting from gradual milling of thin-walled structures. The parameters were selected based on the literature review [2,25,28,35,46,50,53,54]; thus, their choice is a compromise for all analyzed materials. These parameters are optimal for milling polymer composites and aluminum alloys and maximum for milling titanium alloys. Previous studies predominantly analyzed the processing of thin-walled elements made of aluminum and titanium alloys, and the research on thin-walled polymer composites was mainly limited to strength analysis.

## 2. Materials and Methods

The research described in this article was conducted on samples made of aluminum alloy, titanium alloy and carbon fiber-reinforced plastic with epoxy resin. The samples were in the form of cuboidal plates with the initial dimensions of 8 × 15 × 100 mm.

EN-AW-2024 T351 aluminum alloy, in which the main alloying element is copper, is characterized by good machinability, weldability, corrosion resistance and high strength, which makes it popular in the aviation industry. The chemical composition of this alloy and its main properties are given in Table 1.

Ti6Al4V titanium alloy is a material with a slightly higher density than the tested aluminum alloy, but with an excellent strength-to-weight ratio. The chemical composition and characteristic properties of Ti6Al4V are given in Table 2.

The third tested material was carbon fiber-reinforced plastic (CFRP) known under the trade name GR/EP 985-GF-3070. The component that held the entire material together was epoxy resin, constituting 60% of the volume of the entire material. The CFRP material consisted of alternately arranged prepreg layers (at an angle of 90°), which ensured the same strength in all load directions. This arrangement meant that every successive prepreg layer could be rotated by 90° in relation to the previous layer. Composite samples were prepared in a room where the temperature was maintained in the range from 18 °C to 30 °C and the humidity was not higher than 60%. An important factor influencing the “cleanliness” of the composite material was the fact that the quantity of solid particles in the room did not exceed 10,000 per 1 m^3^. To harden the material, the samples were kept for 2 h in an autoclave at a temperature of 177 °C and a pressure of 0.3 MPa. The final stage of sample preparation was a peripheral milling operation. This type of milling is used to obtain samples with precisely defined dimensions as well as surfaces free from sharp and irregular edges. Circular milling also removes burrs and fiber debris. Table 3 shows the properties of the tested carbon fiber-reinforced plastic.

Prepared samples with specified dimensions were subjected to face milling. Face milling was performed using a milling cutter with a diameter of 20 mm. The cutter consisted of a body (20N02R028A20ED10 from Kennametal) and two EDCT10T304PDFR-PCD inserts made of polycrystalline diamond (KD1410). The tools with the insert included and angle of 75° and a clearance angle (minor and major) of 15°.

Milling operations along the x-axis were performed on the Avia-VMC 800 HS vertical machining center with Heidenhain iTNC 530 control. The machining center allows the installation of a Kistler 3D dynamometer (type 9257B) for measuring forces during milling. The dynamometer allows the researcher to measure forces in three directions (X-Y-Z). Values of measured forces are transmitted to the Dynoware data acquisition card (type 5697A) with the DynoWare software (type 2825A) via the connected Kistler 5070 amplifier, which converts electric charge into proportional voltage. Shape errors were measured using a measuring probe supplied with the machining center. The methodology presented in this article can be applied in a different industry. The experimental setups for milling, for elements of the measuring track and for measuring devices are shown in Figure 1. The upper left corner of Figure 1 shows the Avia-VMC 800 HS vertical machining center with Heidenhain iTNC 530 control. On the right side there is an enlarged working area with a place for mounting the cutter, vice and 3D Kistler dynamometer. The measuring probe shown in the Figure 1 on the right was used to measure the deformation after machining. In the lower left corner, the devices for recording the vertical force are presented. The lower right corner of the figure shows a roughness measurement and 3D topography device. The working area of the profilograph is enlarged at the bottom.

Milling was carried out at a constant cutting speed and feed rate while maintaining the depth of cut constant at 1 mm for every tested sample thickness. The experiment involved the face milling of samples with a thickness of 8 mm, 7 mm, 6 mm, 5 mm and 4 mm. Table 4 lists the technological parameters of face milling.

Samples with a length of 100 mm were clamped in a vice put on top of the Kistler dynamometer. The samples were located and clamped so as to allow face milling of an unsupported element with a length of *l* = 50 mm. Figure 2 shows schematically the workpiece fixture. Figure 2 shows the unsupported element length *l*, the thickness *g* and the depth of cut *a_p_*. The position of the cutter in relation to the sample mounting is also shown.

After milling, surface structure geometry, including 3D topography, was examined. The Hommel-Etamic T8000RC 120-140 device was used to develop 3D topography maps and determine surface roughness parameters. The surface topographies with dimensions of 4.8 mm × 4.8 mm presented in this publication were determined with a fixed sampling length of 0.8 mm and an accuracy of 0.01 µm.

For the maximum values of the vertical force, the significance of the influence of element thickness *g* and the length of unsupported element *l* on the obtained results was analyzed. This analysis was performed using Statistica version 13 by analysis of variance (ANOVA).

The first stage of the statistical analysis was to test the normality of distribution using the Shapiro–Wilk test. A significance level of α = 0.05 was adopted for all tests. For the specified level of significance and the number of degrees of freedom, obtained values of the variance of F were compared with the critical value of Fα. After that, the ANOVA was performed and the effect of the levels of the independent variables on the dependent variable was then analyzed by the Tukey test (post hoc test). Before the Tukey test, the homogeneity of variance was tested using the Leven test.

## 3. Results

The study investigated the effect of element thickness *g* and unsupported element length *l* on the maximum values of the vertical force *F_zmax_* for each of the three tested materials. Figures below show the influence of the two variables on the maximum values of the vertical force *F_zmax_* on milling aluminum alloy EN-AW-2024 T351 (Figure 3a), titanium alloy Ti6Al4V (Figure 3b) and carbon fiber-reinforced plastic (CFRP) (Figure 3c).

After milling, the maximum vertical force of the elements made of aluminum alloy is 188 N with the most stable support, where the sample thickness is 8 mm and the distance from the mounting location is only 10 mm. The force values of the aluminum alloy samples decrease with decreasing sample thickness *g* and increasing distance *l*, wherein for the thinnest element and the most distant point the maximum vertical force is 89 N, which is about 50% of the force obtained for the most stable support and the thickest sample. It can be observed that the maximum vertical force decreases dynamically with large distances, which is related to the low value of Young’s modulus, describing material stiffness. The plot also shows a dynamic decrease in this force for the sample thickness of 6 mm and the support distance of 30 mm.

An analysis of the results reveals that the maximum vertical force values are much higher for the titanium alloy elements, and that these values decrease with changing sample thickness and vice-mounting distance. The force values obtained after milling for the titanium alloy elements are the highest values for the sample thickness of 8 mm. A significant difference in the vertical force values can be observed for the elements with thicknesses of 8 mm and 7 mm. With a decrease in the element thickness *g* and an increase in the unsupported distance *l*, the difference in the maximum vertical forces is no longer as great as in the case of the forces obtained for the aluminum alloy elements. This situation results from the Young’s modulus value, which for titanium alloys is 50% higher than the Young’s modulus of aluminum alloys.

The maximum vertical force of the CFRP elements is slightly higher than the force value obtained after machining for the elements made of aluminum alloy. Regarding the CFRP elements, the decrease in the value of this forces is lower compared to that observed for aluminum alloys, which can be explained by the fact that CFRP has a greater Young’s modulus than aluminum alloys. The maximum vertical force decreases, from 208 N after milling the thickest supported element at a distance of 10 mm, to 120 N for the element with a thickness of 4 mm and a distance from the support point of 50 mm. The vertical force decrease is 42% in relation to the initial value of the maximum vertical force.

The study of the normality of distribution confirmed that the examined independent variables had a normal distribution. Therefore, an analysis of variance was performed. Table 5 and Table 6 show the ANOVA results for the maximum values of the vertical force *F_zmax_* obtained for two cases. In the first table, the effect of unsupported element length and constant element thickness on the vertical force value was determined (Table 5), while the other relates to the influence of element thickness and constant unsupported element length on the vertical force value (Table 6).

The analysis results show that the applied variables (unsupported element length and element thickness) significantly affect the obtained values of the maximum vertical force. The proof of this is that for all milling conditions, the probability level *p* is lower than the adopted significance level (α = 0.05), and the value of the test statistic F (3; 16) is higher than the adopted Fα = 3.239.

The results obtained from the ANOVA analysis of variance prove that there are statistical differences in the mean values of the maximum vertical force for the analyzed groups of variables.

Table 7, Table 8 and Table 9 show the Tukey test results for the dependent variables of the maximum vertical forces for the tested materials. The statistical analysis results reveal that there are statistically significant differences between the groups. The conducted post hoc test (Tukey’s test) shows that the change of the unsupported element length and element thickness in the range of the experiment has a statistically significant impact (values less than *p* = 0.05) on the analyzed maximum vertical force.

Another analyzed dependence was the influence of element thickness and unsupported element length on the ratio of vertical force to its average value. For this purpose, the *k*-factor was determined according to Formula (1):*k* = *A*/*S* × 100%,(1)
where *A* is the amplitude of the vertical force value and *S* is the average value of the vertical force. In Figure 4, the example waveform graph of vertical force in milling samples made of aluminum alloy was presented.

The plots in Figure 5a–c show changes in the value of the *k*-factor depending on the element thickness and unsupported element length for the analyzed materials.

For each analyzed material, the ratio between the vertical force amplitude and its average value (*k*-factor) increases with decreasing the thickness *g* and increasing the distance *l*. The change in the *k*-factor value is the most dynamic for the element made of aluminum alloy because this material has a low Young’s modulus value (Figure 5a). For the case of the titanium alloy sample, the change in *k*-factor is small, especially for a large thickness sample. The change in the *k*-factor for the titanium alloy elements is smaller because this material is more rigid, as evidenced by Young’s modulus. The change in the coefficient value observed for the CFRP elements is small because CFRP is very stiff. This material is an alternative to titanium alloys and constitutes a very good replacement for aluminum alloys.

The effect of these variables on surface roughness, namely the Ra parameter, was also investigated. This parameter is most widely used in industry for determining surface roughness. Figure 6a–c show the relationship between sample thickness and unsupported element length and the roughness parameter Ra.

The surface roughness results show that surface roughness increases with decreasing element thickness *g* and increasing unsupported element length *l* for each of the three analyzed materials. The values of the roughness parameter Ra for the elements made of aluminum alloys change very dynamically. For the smallest element thickness and a large distance from the support point, the values of this parameter increase almost two-fold in relation to the value of this parameter obtained for the most stable support and the largest element thickness. The roughness values obtained after machining for the elements made of titanium alloy are higher than the values obtained after machining for the aluminum alloy element because the adopted milling parameters were the upper limit of the cutting conditions for this material. The Ra values for the titanium alloy elements increase significantly less dynamically than for the elements made of aluminum alloys. For the applied milling conditions (thickness *g* = 8 mm and distance *l* = 10 mm), the roughness values increased only by 30%, when compared to the milling process conducted with the thickness *g* = 4 mm and the distance *l* = 50 mm. An analysis of the diagram presented in Figure 6c demonstrates that the roughness Ra values of the CFRP elements are greater than those obtained after milling for the elements made of aluminum and titanium alloys. CFRP is a heterogeneous material characterized by worse surface structure geometry after milling. However, an increase in the length *l* to 50 mm and a decrease in the thickness *g* to 4 mm led to an increase in the Ra value only by 15% compared to the values obtained for the thickness *g* = 8 mm and the length *l* = 10 mm.

An important aspect of milling thin-walled elements is the accuracy of a milled element. This study investigated and illustrated (Figure 7a–c) how the thickness *g* and the unsupported element length *l* affect the *h* value after milling. The *h* value is a difference between the nominal thickness *g* and the thickness obtained after milling *g* + *h*. Positive values mean that the element thickness obtained after milling is greater than the expected nominal value of *g*.

An analysis of the changes (in the nominal thickness of the aluminum alloy elements) demonstrates that the *h* value is maintained in the range from 0.004 mm to 0.130 mm. These values increase with decreasing thickness *g* and increasing distance *l*. The elements made of titanium and CFRP alloys show greater changes in this thickness after milling. The *h* values for the elements made of titanium alloys range from 0.010 mm to 0.265 mm, and for the elements made of CFRP they are from 0.006 to 0.246. Like in the case of the aluminum alloy elements, these values increase with decreasing the thickness *g* and increasing the distance *l*. It would seem that the values obtained for the elements made of titanium alloys and CFRP should show smaller changes in the *h* value compared to the elements made of aluminum alloys. The small deformation of the elements made of aluminum alloys can be explained by this alloy being soft and easy-to-machine material. As a result, the workpiece is milled and not repelled by the tool due to a decreasing stiffness of the workpiece fixture. The cutting resistance of the elements made of aluminum alloy is less significant than Young’s modulus for the elasticity of this material. The greater changes in *h* values for the elements made of titanium alloys and CFRP indicate that the cutting resistance is greater, and therefore that the material is repelled by the tool. These materials belong to the group of difficult-to-machine materials. Although Young’s modulus for titanium alloys and CFRP is almost 50% higher than that for aluminum alloys, it has a less significant effect on the obtained surface than the hardness and material strength of these samples. The plot in Figure 7c also shows the non-linear and non-periodic changes in the value of h, with decreasing thickness *g* and length *l* for CFRP. For the thickness *g* = 6 mm and the length *l* = 50 mm, there is a noticeable change in thickness after machining, its value being greater than the expected nominal value. The change in the thickness *g* and the length *l* plotted in Figure 7b for the elements made of titanium alloys demonstrates that reducing thickness and increasing length led to an almost proportional increase in the thickness *g* + *h* obtained after machining.

The deformation of the elements made of CFRP, as well as aluminum and titanium alloys, are also shown in Figure 8a–c. The deformation maps were generated in the Surfer program for the smallest tested thickness, i.e., *g* = 4 mm. The 15 mm width is marked on one of the horizontal axes, while the unsupported element length from 0 to 50 mm is marked on the other. The vertical axis represents the thickness change *h* in relation to the expected surface.

These maps show that an increase in unsupported element length leads to an increase in the thickness change *h*, which confirms the results presented in the plots in Figure 7a–c.

Apart from the analyzes investigating the influence of element thickness and unsupported element length on the maximum vertical force, *k*-factor, Ra surface roughness and changes in element thickness, 3D topography maps of the surfaces of the tested materials were also prepared. Figure 9a–c show the 3D topography maps with an area of 4.8 mm × 4.8 mm for a milled sample with a thickness of *g* = 4 mm and a length of *l* = 50 mm.

The above maps show the traces of machining for each of the tested materials. On the surface of the elements made of aluminum and titanium alloys, there are noticeable traces after machining. As for the surface of the CFRP element after machining, the characteristic traces forming straight lines or concentric circles are not visible. However, the maximum heights of unevenness obtained after machining for the composite surface are larger, reaching even up to 25 µm, compared to the surface unevenness obtained after milling for the elements made of aluminum and titanium alloys, which is about 18 µm.

## 4. Discussion

The machining of thin-walled elements is predominantly performed in the aviation industry, of which the strength-to-weight ratio of elements is an important aspect. Titanium and aluminum alloys used in the aviation industry are often replaced by composites. Composites have much better strength parameters and, at the same time, low specific weight, and thus—despite high production costs—constitute the most widely used group of materials.

The interest in research on thin-walled elements made of composites and aluminum and titanium alloys has resulted in a series of publications reporting results of studies and analyzes. Regarding the problems related to the machining of elements made of aluminum alloys, there are publications devoted to stress tests [1], measurement of accuracy after machining [23], milling strategies [68], selection of technological parameters [2], as well as various test results describing obtained surface roughness [3,24]. The results obtained in this study for aluminum alloy elements confirm the thesis that reduced element thickness has a negative effect on the deformation of the alloys [24], but they also deny the premise that element thickness does not affect surface roughness [3].

The analysis of machining of elements made of titanium alloys provides information about vibration [4], technological parameters [4] or deformation [69]. This study has confirmed that the support length has a significant influence on element deformation [38].

Previous studies on thin-walled elements made of composite materials have primarily been experimental analyzes and numerical analyses conducted via the finite element method [61,62]. On the other hand, studies related to the assessment of the influence of machining parameters on the machinability of composite materials have been conducted, too [63,64].

The novelty of this study lies in the fact that it focuses on milling thin-walled elements made of carbon fiber-reinforced plastic and aluminum and titanium alloy. The study made it possible to determine the effects of machining and surface quality of elements made of these materials that has great potential for aviation applications. In addition, the results reported in this paper make it possible to draw conclusions from a comparative analysis thanks to the use of the same technological parameters of milling three different materials. This paper presents research of vertical cutting force, the ratio of vertical force amplitude to its average value, surface roughness and accuracy of machined elements, which have not been presented in such a summary so far. Previous studies have only presented conclusions of analyses made for only one material, which makes it impossible to apply them to other materials used under the same conditions in the aviation industry.

## 5. Conclusions

The aim of this work was to investigate machining of thin-walled elements made of aluminum and titanium alloys and polymer composites, as well as to perform a comparative analysis of machining thin-walled structures. Each sample was made entirely of one of three different materials used in the aviation industry. Experiments conducted on elements made of polymer composites and aluminum and titanium alloys provided data about the effect of element thickness *g* and unsupported element length *l* on the maximum vertical force, *k*-factor (a ratio of the vertical force amplitude to its average value), roughness parameter Ra, and surface thickness change after milling with constant technological parameters. The presented methodology has the limitations of examining very thin elements that can break. Moreover, increasing the unsupported length of the element may cause unexpected vibration. ANOVA statistical analysis was also performed.

Based on the study, the following conclusions have been drawn:The maximum vertical force decreases with decreasing thickness of the milled element and increasing unsupported element length. In the case of elements made of aluminum alloys, the decrease in these forces is considerably greater than for the two other types of materials. This relationship can be explained by a low Young’s modulus of this material describing its stiffness, which causes the material to be bent. The less dynamic decrease in forces obtained for CFRP and titanium alloy is due to a 50% higher value of their Young’s moduli.The investigation of *k*-factor, which is a ratio of the force amplitude to the average value of this force, has shown that its value tends to increase with decreasing thickness *g* and increasing length *l*. A clear change in the *k*-factor value for aluminum alloy elements is a consequence of the low Young’s modulus for this material. Slightly smaller changes observed for composite and titanium alloy testify to higher stiffness of these materials, which is confirmed by their greater Young’s moduli.An analysis of the surface roughness of thin-walled structures after milling has shown that, for aluminum alloy elements, the value of this parameter almost doubled when comparing two extreme machining conditions. This has also been observed for the use of the maximum values of milling parameters in quite high roughness values of titanium alloy. However, the change in the roughness parameters of both the composite material and titanium alloy is not as significant as that obtained for aluminum alloy.Interesting conclusions result from an analysis of the change in thickness of thin-walled elements after milling. It has been shown that the elements made of aluminum alloy, despite the low Young’s modulus of this alloy, are characterized by relatively small changes in their shape. This phenomenon may be explained by good machinability of aluminum alloys in combination with their low hardness, which allowed for lower force *F_zmax_* on the sharp edges of the milling cutter. A post-machining analysis of the surfaces of the composite material and titanium alloy, characterized by a high Young’s modulus value as well as high hardness and strength, has shown that the changes in element thickness were higher. Greater changes in element thickness from the expected nominal thickness were caused by the material being repelled from the tool. The difficult-to-machine properties of these materials have been found to have a more significant influence on deformation than the high Young’s modulus value.

Based on the literature review and the results obtained in this study, it can be concluded that the investigation of thin-walled structures, particularly those made of composite materials, is an area for further research. We plan to analyze the influence of variable technological parameters on the maximum vertical force, the *k*-factor, surface roughness and accuracy of machined elements. It is planned to study the machining of thin-walled structures using a high-speed camera. The aim of this study will be to observe the behavior of the material during machining. In addition, studies on finite element analysis are planned.

## Figures and Tables

**Figure 1 materials-15-05949-f001:**
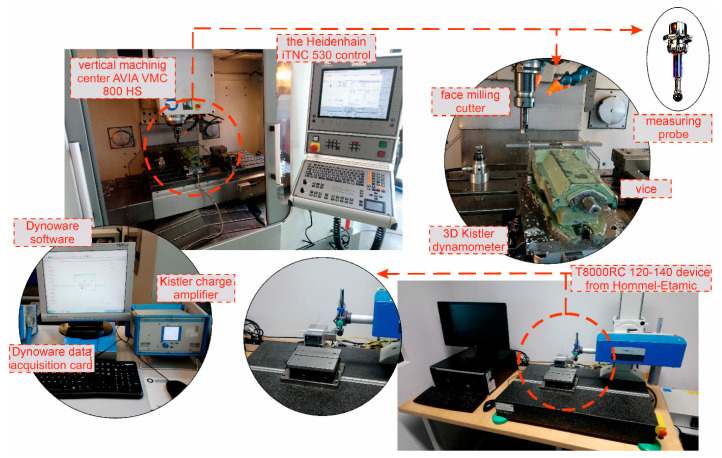
Experimental setup for milling, elements of the measuring track and measuring devices.

**Figure 2 materials-15-05949-f002:**
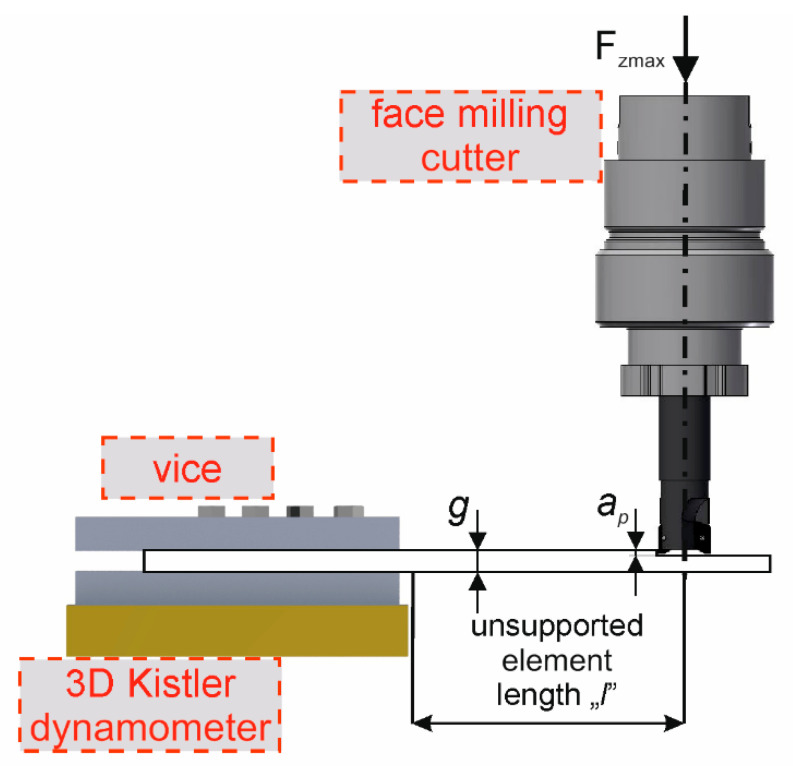
Scheme showing the fixture of the workpiece.

**Figure 3 materials-15-05949-f003:**
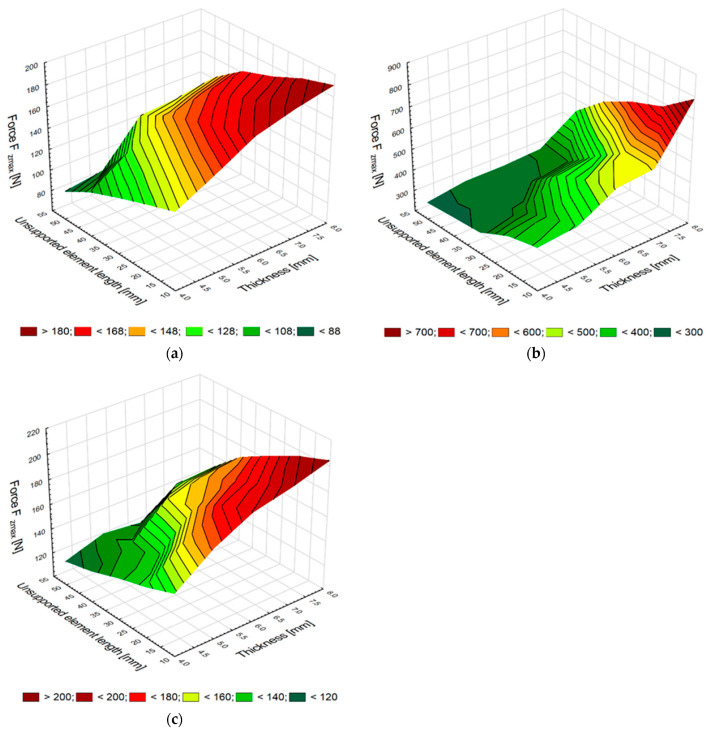
Element thickness and unsupported element length versus maximum vertical force obtained in milling samples made of aluminum alloy (**a**) and titanium alloy (**b**) and CFRP (**c**).

**Figure 4 materials-15-05949-f004:**
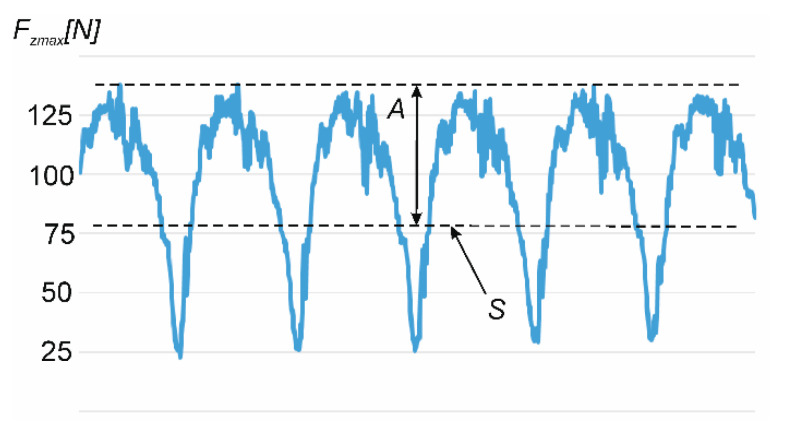
The example waveform graph of vertical force in milling samples made of aluminum alloy.

**Figure 5 materials-15-05949-f005:**
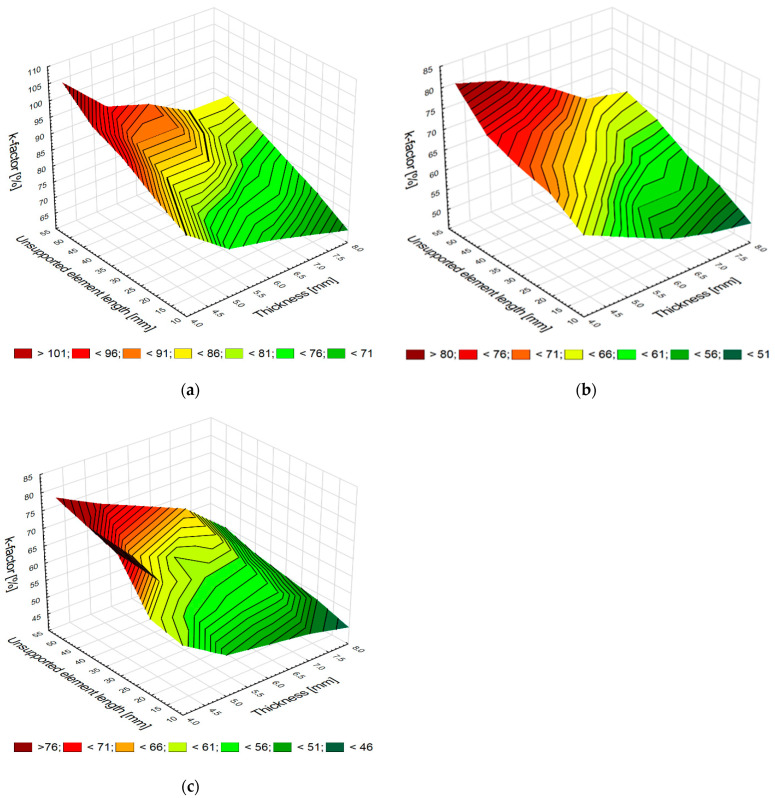
Element thickness and unsupported element length versus the *k*-factor values in milling samples made of aluminum alloy (**a**) and titanium alloy (**b**) and CFRP (**c**).

**Figure 6 materials-15-05949-f006:**
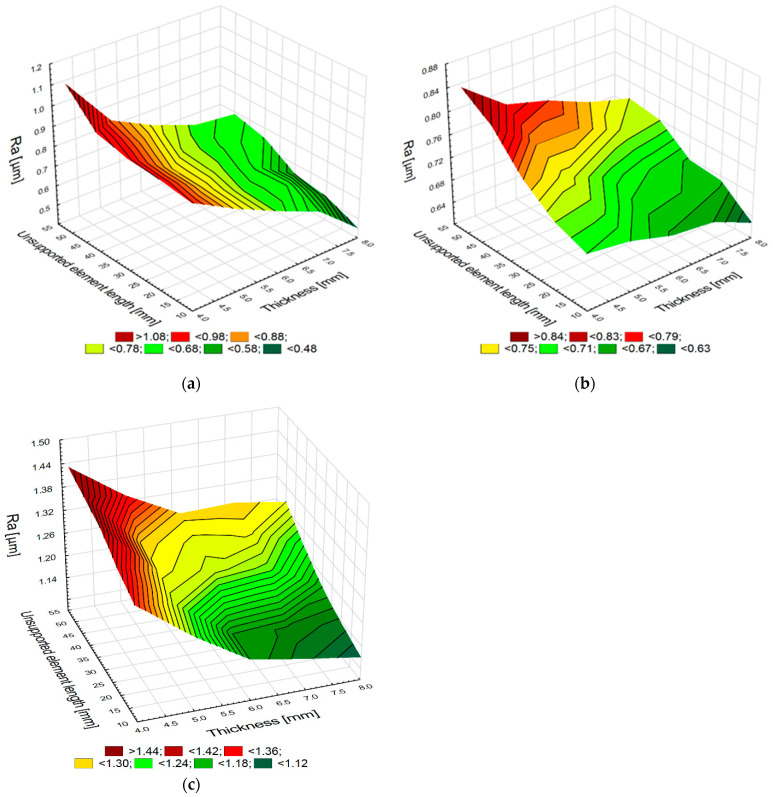
Element thickness and unsupported element length versus the Ra parameter values in milling samples made of aluminum alloy (**a**) and titanium alloy (**b**) and CFRP (**c**).

**Figure 7 materials-15-05949-f007:**
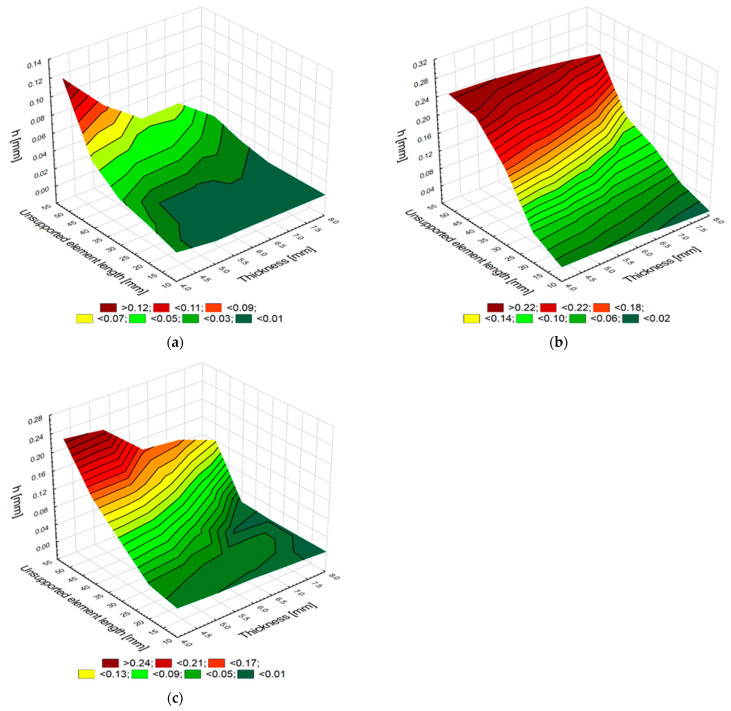
Element thickness and unsupported element length versus the thickness *h* value after milling samples made of aluminum alloy (**a**) and titanium alloy (**b**); and CFRP (**c**).

**Figure 8 materials-15-05949-f008:**
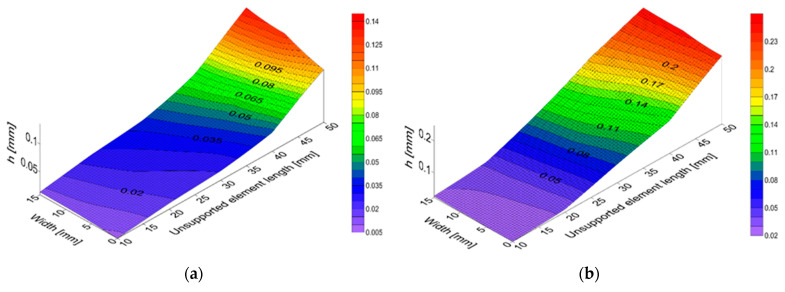
Thickness change *h* obtained after milling for the samples of aluminum alloy (**a**) and titanium alloy (**b**) and CFRP (**c**) with a thickness of 4 mm.

**Figure 9 materials-15-05949-f009:**
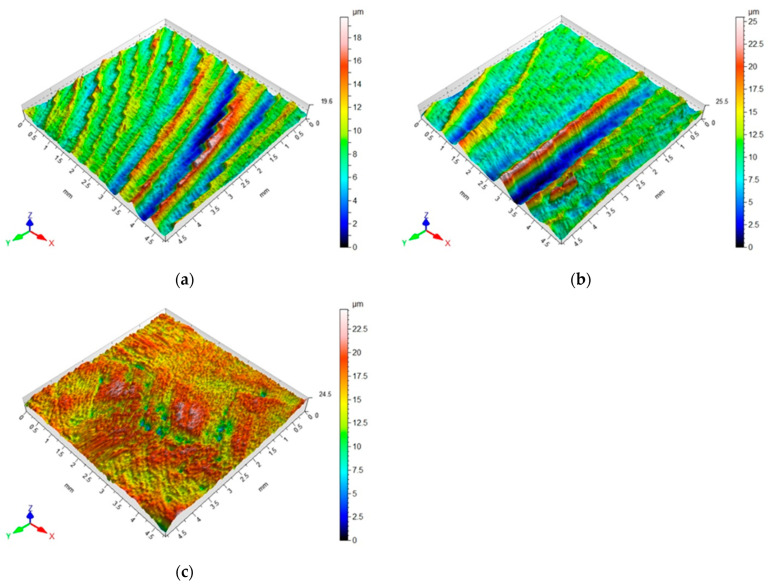
3D topography maps obtained after milling for the samples made of aluminum alloy (**a**) and titanium alloy (**b**) and CFRP (**c**) with thickness *g* = 4 mm and length *l* = 50 mm.

**Table 1 materials-15-05949-t001:** Chemical composition and main properties of EN-AW-2024 T351 aluminum alloy [5,65,66].

Chemical Composition [%]					
Cu	Mg	Fe	Si	Mn	
3.8–4.9	1.2–1.8	≤0.5	≤0.5	0.3–0.9	
Zn	Zr + Ti	Ti	Cr	other	Al
≤0.25	≤0.2	≤0.15	≤0.1	0.2	rest
**Properties**					
Density [g/cm^3^]				2.78	
Tensile strength [MPa]				460	
Young’s modulus [GPa]				73	
Hardness [HB]				120	

**Table 2 materials-15-05949-t002:** Chemical composition and main properties of Ti6Al4V titanium alloy [29].

Chemical Composition [%]							
Al	V	Fe	C	N	H	other	Ti
5.5–6.5	3.5–4.5	≤0.25	≤0.08	≤0.03	≤0.013	≤0.4	rest
**Properties**							
Density [g/cm^3^]					4.43		
Tensile strength [MPa]					930		
Young’s modulus [GPa]					110		
Hardness [HB]					334		

**Table 3 materials-15-05949-t003:** Properties of carbon fiber-reinforced plastic [67].

Properties	
Density [g/cm^3^]	1.6
Tensile strength [MPa]	1500
Young’s modulus [GPa]	135

**Table 4 materials-15-05949-t004:** Technological parameters of face milling.

Cutting Speed *v_c_* [m/min]	Feed Rate *v_f_* [mm/min]	Depth of Cut *a_p_* [mm]	Thickness of Sample *g* [mm]
100	478	1	4–8

**Table 5 materials-15-05949-t005:** ANOVA results for the maximum vertical force versus unsupported element length and constant element thickness.

	*F_zmax_*				
	DF	SS	MS	F	*p*
**Al**	4	7170.4	1792.5	497.5	0.0000
**Ti**	4	339,186	84,797	10,402	0.0000
**CFRP**	4	9211.8	2302.9	321.9	0.0000

**Table 6 materials-15-05949-t006:** ANOVA results for the maximum vertical force versus element thickness and constant unsupported element length.

	*F_zmax_*				
	DF	SS	MS	F	*p*
**Al**	4	10,622.8	2655.7	670.4	0.0000
**Ti**	4	473,058	118,264	18,354	0.0000
**CFRP**	4	8901.4	2225.4	270.9	0.0000

**Table 7 materials-15-05949-t007:** Comparative analysis of the significance of differences (post hoc—Tukey test) between the maximum values of vertical forces obtained for aluminum alloy elements in milling with different thicknesses and distances.

** *F_zmax_* **					
** *g* **					
** *g* **	10	20	30	40	50
**10**		0.000142	0.000132	0.000132	0.000132
**20**	0.000142		0.000132	0.000132	0.000132
**30**	0.000132	0.000132		0.000133	0.000132
**40**	0.000132	0.000132	0.000133		0.000132
**50**	0.000132	0.000132	0.000132	0.000132	
** *F_zmax_* **					
** *l* **					
** *l* **	8	7	6	5	4
**8**		0.000174	0.000132	0.000132	0.000132
**7**	0.000174		0.000132	0.000132	0.000132
**6**	0.000132	0.000132		0.000133	0.000132
**5**	0.000132	0.000132	0.000133		0.000132
**4**	0.000132	0.000132	0.000132	0.000132	

**Table 8 materials-15-05949-t008:** Comparative analysis of the significance of differences (post hoc—Tukey test) between the maximum values of vertical forces obtained for titanium alloy elements in milling with different thicknesses and distances.

** *F_zmax_* **					
** *g* **					
** *g* **	10	20	30	40	50
**10**		0.000132	0.000132	0.000132	0.000132
**20**	0.000132		0.000132	0.000132	0.000132
**30**	0.000132	0.000132		0.000132	0.000132
**40**	0.000132	0.000132	0.000132		0.000132
**50**	0.000132	0.000132	0.000132	0.000132	
** *F_zmax_* **					
** *l* **					
** *l* **	8	7	6	5	4
**8**		0.000132	0.000132	0.000132	0.000132
**7**	0.000132		0.000132	0.000132	0.000132
**6**	0.000132	0.000132		0.000132	0.000132
**5**	0.000132	0.000132	0.000132		0.000132
**4**	0.000132	0.000132	0.000132	0.000132	

**Table 9 materials-15-05949-t009:** Comparative analysis of the significance of differences (post hoc—Tukey test) between the maximum values of vertical forces obtained for CFRP elements in milling with different thicknesses and distances.

** *F_zmax_* **					
** *g* **					
** *g* **	10	20	30	40	50
**10**		0.000497	0.000132	0.000132	0.000132
**20**	0.000497		0.000135	0.000132	0.000132
**30**	0.000132	0.000135		0.000132	0.000132
**40**	0.000132	0.000132	0.000132		0.000132
**50**	0.000132	0.000132	0.000132	0.000132	
** *F_zmax_* **					
** *l* **					
** *l* **	8	7	6	5	4
**8**		0.000731	0.000132	0.000132	0.000132
**7**	0.000731		0.008440	0.000132	0.000132
**6**	0.000132	0.008440		0.000132	0.000132
**5**	0.000132	0.000132	0.000132		0.000132
**4**	0.000132	0.000132	0.000132	0.000132	

## Data Availability

Not applicable.

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
