# Peer review of "Milling of Three Types of Thin-Walled Elements Made of Polymer Composite and Titanium and Aluminum Alloys Used in the Aviation Industry"

_materials, 2022, doi:10.3390/ma15175949_

Round 1

Reviewer 1 Report

The overall quality of the submitted manuscript is satisfactory. Results of the conducted research are good. The entirety of the work is investigating face milling thin-walled elements made of titanium alloy, aluminum alloy and polymer composite.

1.     An abstract needs quantitative results (percentage increment or decrement results).

  1. There are a lot of figures in the manuscript. Combine figures are recommended for good readability. 

  1. Image 1 should be redesigned, some pictures in the figure are not of good quality for example (3d topography).

  1. There is a typo error in table 4 i.e. (thickness 4-8 mm). These types of errors are observed in the whole manuscript. Please check the overall manuscript before submitting of revision.

  1. The authors reported a lot of results in the manuscript 

For example Pg-10, line-347-352

The analysis results show that the applied variables significantly affect the obtained values of the vertical force. For all milling conditions, the probability level p is lower than the adopted significance level (α = 0.05), and the value of the test statistic F (3; 16) is lower than the adopted Fα = 3.239.

The results obtained from the ANOVA analysis of variance prove that there are statistical differences in the mean values of the maximum vertical force for the analyzed groups of variables.

Kindly explain the valid reason for these types of results.

6.     Furthermore, a description of the figures is required.

‘Kindly correct all such mistakes in the manuscript before uploading the revised version.’

Author Response

We would like to thank the Editor for their consideration, and the Reviewers for the time spent on carefully reviewing this work and for their valuable deep insight and comments. We feel that this paper is now clearer, more thoroughly discussed and better-referenced.

The work has been revised to address the reviewers’ suggestions. In attachment  find hereafter a point-by-point reply to the comments and suggestions. Any revisions to the manuscript was marked up using the “Track Changes”.

Reviewer 2 Report

This manuscript conducts the research on face milling of thin-walled elements made of titanium alloy, aluminum alloy and polymer composite. The effect of element thickness g and unsupported element length l on the maximum vertical force, k-factor (a ratio of the vertical force amplitude to its average value), surface roughness Ra, and surface thickness change and surface 3D topography map after milling with constant technological parameters are presented graphically. Some suggestions are as follows:

1. The title of the manuscript is easy to cause ambiguity. The thin-walled elements seems like an sandwich structure made of polymer composite, titanium and aluminum alloys.

2. The geometric parameters of the tool need to be given in the manuscript.

3. The introduction is too long. Some of the content is not related to the research topic.

4. It is suggested to give the waveform graph of vertical force to illustrate the k-factor.

5. It is suggested to only choose titanium alloy and aluminum alloy as research object, focusing on the effects of the vertical force and structural stiffness on the milling deformation of the thin-walled elements.

Author Response

(The authors gave the same response as above.)

Reviewer 3 Report

This article presented a study of investigating the cutting force and surface topography in cutting thin-wall materials.

This is an interesting field, but some issues need to be noted:

1. In the title, “thin-walled elements made of polymer composite and titanium and aluminum alloys” is not a good expression. Meanwhile, In Line 526, “the workpiece is a thin-walled structure made of three different materials”. This is confusing.

It looks like the workpiece is a composite consisting of titanium alloy, aluminum alloy, and CFRP rather than three individual materials. Please comment on that.  

2. The authors need to further present the innovation of this work. For example, in Line 515, “The novelty of this study lies in the fact that it focuses on milling thin-walled elements made of carbon fiber reinforced plastic”. The authors need to supplement the innovation of milling the other materials.

3. The authors must double-check the article writing, and lots of obvious grammar problems can be found throughout the article. 

For example: as shown in the Abstract, “causes may problems, which creates” should be “causes may problems, which create”. 

In Line 478, it should be Figs. 8a-c.

4. The authors need to shorten the Introduction. Some irrelevant references are cited.

5. The authors need to pay attention to the present study format. For example, Fig. 1 is not clear.

Author Response

(The authors gave the same response as above.)

Reviewer 4 Report

Comments to improve the manuscript are as follows:

1. The universality of the methodology is currently hidden. Why do the authors think that this methodology can only be applied in the aviation industry? I think it should not be limited to the aviation industry.

2. Additionally, the term "thin-walled elements" is used throughout the manuscript. When we can say that an element is thin-walled and when it is not. Where is the border?

3. Given that three different materials are machining, how did you select the milling cutter?

4. How are the milling parameters (cutting speed, feed rate, depth of cut) shown in table 4 selected? The authors state in the manuscript the following "The parameters were selected based on the literature review, thus their choice is a compromise for all analyzed materials" However, the literature is not cited. Elaborate further on how the compromise was implemented. Be detailed.

5. Why are the milling parameters (cutting speed, feed rate, depth of cut) shown in table 4 not varied?

6. I think that instead of the term "fastened" it is more appropriate to use "locating and clamping".

7. Why is it an unsupported element with a length of l = 50 mm. Why is unsupported element length not varied? Why is 50mm representative?

8. Write in the Conclusions section the limitations of your methodology.

9. The following sentence is very general "Based on the literature review and the results obtained in this study, it can be concluded that the investigation of thin-walled structures, particularly those made of composite materials, is an area for further research." Write specifically what future research will be conducted.

Author Response

(The authors gave the same response as above.)

Round 2

Reviewer 2 Report

The introduction is too long. Some of the content is not related to the research topic. Suggest to reduce reasonably.

Author Response

We would like to thank the Editor for their consideration, and the Reviewer for the time spent on carefully reviewing this work and for their valuable deep insight and comments. We feel that this paper is now clearer, more thoroughly discussed and better-referenced. The work has been revised to address the reviewer suggestion. Any revisions to the manuscript was marked up using the “Track Changes”.

Reply to the comment in the attachment.

Reviewer 3 Report

This article presented a study of investigating the cutting force and surface topography in cutting thin-wall materials.

Accepted as it is.

Author Response

We would like to thank the Editor for their consideration, and the Reviewer for the time spent on carefully reviewing this work.

Reply to the comment in the attachment.

Reviewer 4 Report

The manuscript has been corrected.

Author Response

(The authors gave the same response as above.)
